# Transmitted HIV Drug Resistance in Bulgaria Occurs in Clusters of Individuals from Different Transmission Groups and Various Subtypes (2012–2020)

**DOI:** 10.3390/v15040941

**Published:** 2023-04-10

**Authors:** Ivailo Alexiev, Anupama Shankar, Yi Pan, Lyubomira Grigorova, Alexandra Partsuneva, Reneta Dimitrova, Anna Gancheva, Asya Kostadinova, Ivaylo Elenkov, Nina Yancheva, Rusina Grozdeva, Dimitar Strashimirov, Mariana Stoycheva, Ivan Baltadzhiev, Tsetsa Doichinova, Lilia Pekova, Minas Kosmidis, Radoslava Emilova, Maria Nikolova, William M. Switzer

**Affiliations:** 1National Reference Laboratory of HIV, National Center of Infectious and Parasitic Diseases (NCIPD), 1504 Sofia, Bulgaria; 2Division of HIV Prevention, National Center for HIV, Viral Hepatitis, STD, and TB Prevention, Centers for Disease Control and Prevention, Atlanta, GA 30329, USA; 3Specialized Hospital for Active Treatment of Infectious & Parasitic Diseases, 1606 Sofia, Bulgaria; 4Department of Infectious Diseases, Medical University, 4002 Plovdiv, Bulgaria; 5Department of Infectious Diseases, Medical University, 5800 Pleven, Bulgaria; 6Clinic of Infectious Diseases, Medical University, 6000 Stara Zagora, Bulgaria; 7Clinic of Infectious Diseases, Medical University, 9002 Varna, Bulgaria; 8National Reference Laboratory of Immunology, National Center of Infectious and Parasitic Diseases (NCIPD), 1504 Sofia, Bulgaria

**Keywords:** HIV-1, transmitted drug resistance, transmission clusters, subtypes, male-to-male sexual contact

## Abstract

Transmitted HIV drug resistance in Bulgaria was first reported in 2015 using data from 1988–2011. We determined the prevalence of surveillance drug resistance mutations (SDRMs) and HIV-1 genetic diversity in Bulgaria during 2012–2020 using polymerase sequences from 1053 of 2010 (52.4%) antiretroviral therapy (ART)-naive individuals. Sequences were analyzed for DRM using the WHO HIV SDRM list implemented in the calculated population resistance tool at Stanford University. Genetic diversity was inferred using automated subtyping tools and phylogenetics. Cluster detection and characterization was performed using MicrobeTrace. The overall rate of SDRMs was 5.7% (60/1053), with 2.2% having resistance to nucleoside reverse transcriptase inhibitors (NRTIs), 1.8% to non-nucleoside reverse transcriptase inhibitors (NNRTIs), 2.1% to protease inhibitors (PIs), and 0.4% with dual-class SDRMs. We found high HIV-1 diversity, with the majority being subtype B (60.4%), followed by F1 (6.9%), CRF02_AG (5.2%), A1 (3.7%), CRF12_BF (0.8%), and other subtypes and recombinant forms (23%). Most (34/60, 56.7%) of the SDRMs were present in transmission clusters of different subtypes composed mostly of male-to-male sexual contact (MMSC), including a 14-member cluster of subtype B sequences from 12 MMSC and two males reporting heterosexual contact; 13 had the L90M PI mutation and one had the T215S NRTI SDRM. We found a low SDRM prevalence amid high HIV-1 diversity among ART-naive patients in Bulgaria during 2012–2020. The majority of SDRMs were found in transmission clusters containing MMSC, indicative of onward spread of SDRM in drug-naive individuals. Our study provides valuable information on the transmission dynamics of HIV drug resistance in the context of high genetic diversity in Bulgaria, for the development of enhanced prevention strategies to end the epidemic.

## 1. Introduction

Antiretroviral therapy (ART) has decreased morbidity and mortality in HIV-1-infected individuals and is the essential component of the UNAIDS initiative to end the HIV epidemic by reducing HIV-1 transmission [1,2]. However, the emergence of HIV drug resistance can compromise the effectiveness of antiretroviral drugs in reducing HIV incidence and HIV-associated morbidity and mortality [3]. HIV drug resistance mutations (DRMs) can be transmitted to newly infected individuals and can adversely affect first-line antiretroviral therapy regimens and pre-exposure prophylaxis (PrEP) [4]. In this regard, the World Health Organization (WHO), the International AIDS Society–USA, and the European AIDS Clinical Society (EACS) guidelines recommend HIV drug resistance testing for drug-naive patients before beginning ART [3,5].

In 2009, an expert panel generated a list of nonpolymorphic HIV-1 DRMs to monitor transmitted drug resistance (TDR) [6,7]. This surveillance DRM (SDRM) list contains 93 DRMs, including 34 nucleoside reverse transcriptase inhibitor (NRTI)-associated DRMs at 15 reverse transcriptase (RT) positions, 19 non-nucleoside reverse transcriptase inhibitor (NNRTI)-associated DRMs at 10 RT positions and 40 protease inhibitor (PI)-associated DRMs at 18 positions. The SDRM list was widely adopted by the WHO and research institutions and has been used to standardize monitoring and facilitate comparisons of TDR between different studies and countries [8]. TDR can differ in different regions of the world and can change over time as new ART becomes available. For example, the prevalence of TDR between 2014 and 2019 was 14.2% in North America, followed by 9.1% in Latin America and the Caribbean, 8.5% in Europe, 6.0% in sub-Saharan Africa, and 4.1% in Southeast Asia [9,10]. In the Balkan countries, TDR varies widely, with 16.4% reported in Croatia, 7.8% in Greece, 14.8% in Romania, 8.8% in Serbia, and 2.4% in Slovenia [11,12,13,14,15]. Rhee et al. recently reported that NNRTI resistance has increased globally since 2013 in ART-naive adults [10].

Bulgaria has a relatively low HIV prevalence of less than 0.1% compared to its neighbors and, as of 2020, a total of 3474 people have been diagnosed with HIV. However, we have previously reported a wide diversity of HIV-1 subtypes in Bulgaria, including circulating recombinant forms (CRFs) and unique recombinant forms (URFs), which can complicate our understanding of HIV-1 epidemiology and interpretation of DRMs [16,17,18]. Following multiple introductions of different HIV-1 genotypes from abroad into Bulgaria and random founder events, numerous HIV-1 strains then spread unevenly into different transmission risk groups [19], with subtype B infection found mostly in those reporting male–male sexual contact (MMSC). CRF01_AE and CRF02_AG subtypes were seen among two geographically distinct subgroups of people who inject drugs (PWIDs), and the greatest diversity was identified in people reporting only heterosexual (HET) infection [20,21,22,23].

Following European and Bulgarian national guidelines, new HIV diagnoses in Bulgaria are tested for antiretroviral resistance before starting ART. However, little is known about TDR in Bulgaria, with baseline results published in 2015 using data from 1988–2011 [17]. In the 2015 report, we documented a 5.2% SDRM prevalence, with most (3.6%) resistance to NRTIs, followed by NNRTIs (1.6%) and PIs (0.9%). Most SDRMs were in heterosexuals (62.5%), followed by MMSC (25%), and PWIDs (12.5%). SDRMs were present in seven different HIV-1 subtypes, with only two subtype B sequences with SDRMs from a heterosexual and an MMSC clustering together by phylogenetic analysis. Since 2011, treatments have changed, new antiretroviral drugs have become available, and HIV-1 continues to spread in Bulgaria, with the identification of an additional 2010 new HIV-1 infections in ART-naive individuals. Assessment of SDRMs and viral diversity is a key factor in monitoring the HIV-1 epidemic and optimizing first-line therapy for long-term management of HIV-1 infection in Bulgaria. Our study aims to determine the prevalence and characterize the phylodynamics of SDRMs in Bulgaria using newly developed bioinformatics tools. Our findings will help to better focus public health intervention efforts for effectively controlling ongoing spread and stopping the HIV-1 epidemic in Bulgaria.

## 2. Materials and Methods

### 2.1. Study Design and Specimen Preparation

In this nationwide study, all available blood samples from ART-naive individuals diagnosed with HIV infection in Bulgaria between 2012 and 2020 were tested for TDR before starting ART. Epidemiological and demographic data were collected during patient self-assessment interviews following national regulations that were provided by the clinics responsible for the management of HIV-1 patients in six major cities in Bulgaria (Sofia, Plovdiv, Pleven, Stara Zagora, and Varna). Blood samples were linked to demographic and clinical data by using anonymous numerical codes, in accordance with the ethical standards of Bulgaria as previously described [16]. Plasma samples were prepared at the National HIV Reference Laboratory as previously described and stored at −80 °C until tested [16]. Viral RNA was isolated from plasma samples using the Abbott Viroseq HIV-1 Genotyping Test and/or the QIAmp Viral RNA Mini Kit (Qiagen, Hilden, Germany).

### 2.2. Sequence Analyses

The HIV-1 protease and part of the RT region of the polymerase (*pol*) gene were generated using either the Viroseq HIV-1 Genotyping Test (Abbott, Chicago, IL, USA), the Applied Biosystems 3130xl genetic analyzer or the TruGene DNA Sequencing System (Siemens Healthcare, Erlangen, Germany), or the OpenGene DNA sequencing system, as previously described [23].

HIV-1 subtypes were first inferred by using the automated subtype tool COMET v2.4 [24], the REGA HIV-1 subtyping tool v3.0 [25], the jumping profile Hidden Markov Model (jpHMM) [26], and then by phylogenetic analysis. Sequences were aligned using MAFFT v7 [27,28] followed by manual editing with AliView v1.23 [29]. The final alignment length was 916 nucleotides. Phylogenetic relationships were inferred using the approximate maximum likelihood (ML) method implemented in FastTree v2.1.7 (FML) and the general time reversible nucleotide substitution was model inferred using ML-based model selection in MEGA v7.0.26 [30,31]. Clades in the phylogenetic trees were defined by Shimodaira–Hasegawa (SH) support values > 0.7 inferred in FML. For phylogenetic-based subtyping, we included curated global reference sequences (n = 3812) from the Los Alamos National Laboratory database (LANL) [32]. Final subtyping calls were based on a combination of results from subtyping, phylogenetic and recombination analyses. For example, if an automated subtyping tool identified a potential URF but sequences belonged to a different phylogenetic clade, then additional recombination and phylogenetic analyses were conducted. Confirmation of the genetic composition of some novel URFs required additional FML phylogenetic analyses using potential recombinant *pol* fragments with breakpoint positions determined by jpHMM and pre-made, non-recombinant group M subtype reference sequences available at LANL. FigTree v1.4.4 was used to visualize the phylogenetic trees [33].

HIV-1 SDRMs were determined with the Calibrated Population Resistance (CPR) Tool version 8.1 (https://hivdb.stanford.edu/cpr/) (accessed on 10 May 2023) that uses the 2009 WHO SDRMs list [34]. CPR utilizes the Stanford University HIV Drug Resistance Database and Sierra algorithm for inference of drug resistance.

We inferred and explored HIV-1 *pol* genetic transmission networks in MicrobeTrace by using a Tamura–Nei genetic distance (*d*) cutoff of 1.5% [35]. This cutoff has been previously used to characterize established and recent HIV-1 transmission dynamics in Bulgaria [22,23]. We used molecular as well as demographic data to examine the distribution of population characteristics across transmission clusters in the network and any potential association with TDR. In order to do this, nodes were colored by subtype and shaped by DRM type and links were colored by transmission risk category.

A 10% subset of anonymous HIV-1 *pol* sequences from our study were deposited in GenBank to help protect the privacy of the participants. The accession numbers are MN978144—MN978148, MN978150—MN978155, MN978157—MN978159, MN978161—MN978184, MN978188—MN978204, MN978206—MN978212, MN978214—MN978217, MN978220—MN978237, MN978242—MN978252, MN978254—MN978264. The authors can be contacted for additional data as appropriate.

### 2.3. Statistical Analysis

Potential associations of the presence and absence of SDRMs with selected epidemiological characteristics such as gender, age, country of origin, likely country of infection, subtype, and transmission risk were evaluated statistically using a univariate analysis with Fisher’s exact test in SAS 9.4. For determining potential associations of SDRM within the subtype classification, we used a Bonferroni correction, and the significance level was adjusted to 0.01 (=0.05/5) for the five variables in the subtype category.

## 3. Results

### 3.1. Study Population

A total of 1053 (52.4%) of 2010 ART-naive patients diagnosed with HIV-1 in Bulgaria during 2012–2020 participated in this study. The median age of the participants at the time of the diagnosis was 36.0 years. Among the total number of HIV-1 infections with SDRMs, the majority (90.0%) were male and potential infection routes included MMSC (60.0%), HET (36.6%), PWID (3.3%) and MMSC/PWID (0%) (Table 1). Most of the patients (80.0%) reported that they likely acquired HIV infection in Bulgaria, while 20.0% indicated they were likely infected outside Bulgaria. People infected abroad were considered migrants and comprised Bulgarians likely infected while traveling or living abroad and foreigners who were diagnosed during their visit to Bulgaria. Of the demographic and genotype characteristics in HIV-1 infected people with SDRMs, only two showed an association with mutations (Table 1). Subtype was associated with the presence of SDRMs, while being a blood donor was associated with the absence of SDRMs (Table 1). Additional statistical analysis showed that subtype CRF12_BF was significantly associated with SDRMs compared to the reference subtype B (*p* < 0.001).

### 3.2. High HIV-1 Diversity in Drug-Naive Patients in Bulgaria

Subtyping and phylogenetic analysis showed the presence of highly diverse HIV in the population, with the majority being subtype B (n = 636) (Table 1), followed by 120 of CRF01_AE, 73 of subtype F1, 55 of CRF02_AG, 39 of A1, 33 of A6, 32 of URF F1B (described below), 13 of subtype C, 13 of URFs, eight of CRF12_BF, five of URF BG, and 13 other subtypes with four people or fewer in each type (Table 2). URFs were defined as sequences located between known reference sequences in the phylogeny with support from automated subtyping tools. When large clades of novel URFs containing at least 15 sequences were identified, we performed additional sequence analyses to further characterize these variants. REGA and COMET identified most of these as B and F1 recombinants, but some were also identified as F1/B/D recombinants by COMET (n = 14) and others were classified as subtype B by REGA (n = 5). Sequence analysis using jpHMM identified these 32 sequences as F1 and B unique recombinants, all with the same breakpoints. To further clarify the genetic composition and breakpoints of these F1B URFs, we conducted ML phylogenetic analysis that included group M reference sequences. From positions 2253–2652 in the alignment, the 32 sequences formed a clade with strong support (SH = 0.84) with the subtype F1 references, while from positions 2653–3554 they all formed a clade with the subtype B references with strong support (SH = 0.93).

Additional characteristics of the drug-naive study population by subtype classification are provided in Table 2, including gender, potential infection location, transmission risk, and SDRMs.

### 3.3. Evidence for Transmitted Drug Resistance in Drug-Naive Patients

The overall rate of SDRMs in the study population was 5.7% (60/1053), of which most were men (54/60, 90.0%) (Table 1). The prevalence of SDRMs was highest among MMSC (36/60, 60.0%), followed by HET (22/60, 36.6%) and PWID (2/60, 3.3%).

SDRMs were found in 20.0% (12/60) of patients who reported most likely acquiring HIV-1 infection while traveling or living abroad (Table 1). SDRMs were also detected in patients with other STI co-infections (15/60, 25.0%), contact partners (3/60, 5.0%), sex workers (2/60, 3.3%), pregnant women (2/60, 3.3%), and blood donors (2/60, 3.3%). The highest number of SDRMs were found in individuals infected with HIV-1 subtype B (32/60, 53.3%), which is the most abundant subtype in the study population (59.5%) (Table 1). TDRMs were also identified in individuals infected with other subtypes, including subtype A1 (5/60, 8.3%), F1 (4/60, 6.7%), CRF02_AG (4/60, 6.7%), CRF12_BF (5/60, 8.3%), and 10 (16.7%) other subtypes or recombinant forms. Statistical analysis of SDRMs by year was not significant (*p* = 0.3345), likely because of the sparseness of the data.

Sequences from 23/60 people (38.3%) showed resistance to NRTIs, 19 (31.8%) showed resistance to NNRTIs and 22 (36.7%) showed resistance to PIs. The two most common NRTI mutations were M184V and M41L, followed by another 16 mutations, with 10 patients having multiple NRTI SDRMs (Figure 1A). The most common NNRTI mutation was K103N, which was identified in 15 people (25%), followed by five other mutations. The most common PI mutation was L90M, identified in 14 patients (23.3%), followed by seven other mutations. Fifty-six of the 60 patients (93.3%) had single-class SDRMs, while dual-class SDRMs (NRTI + NNRTI or NRTI + PI) were identified in four people (6.7%), of which three were men (two MMSC and one HET) and one was a HET woman.

Four levels of inferred phenotypic drug resistance were observed among different antiretroviral drugs, as shown in Figure 1B. The highest level of phenotypic resistance for NRTIs was seen with abacavir (ABC), lamivudine (3TC) and emtricitabine (FTC), efavirenz (EFV), nevirapine (NVP) and rilpivirine (RPV) for the NNRTIs, and to the PI nelfinavir (NFV). Intermediate, low-level and potential low-level resistance was also inferred for various drugs from each of the three antiretroviral classes.

Overall, our results show a low level of SDRMs in Bulgaria, occurring in various transmission groups and HIV-1 subtypes with resistance different antiretroviral drug classes.

### 3.4. Detection of HIV-1 Transmission Clusters with SDRMs

We next determined the number and characteristics of transmission clusters in our study population using MicrobeTrace. 

#### 3.4.1. Subtype and Prevalent Infection Composition of Clusters

Transmission network analyses at d = 1.5% identified a total of 78 clusters with sizes ranging from two (n = 37) to 102 (Figure 2). The two largest clusters contained 100 and 97 subtype B sequences and the majority of the clusters (56/78, 72%) were also subtype B. We also detected eight CRF01_AE clusters ranging in size from two to 37 members, three subtype F1 clusters with two to 63 members, three subtype A1 clusters with three to 18 members, two CRF02_AG clusters with three and 27 members, two small subtype A6 clusters with two and three members, one four-member cluster of CRF12_BF sequences, and one large 25-member cluster of novel F1B URFs. These results highlight further the high HIV-1 genetic diversity that is spreading in Bulgaria.

We also colored links in clusters by transmission risk to measure the eight different types of potential transmission occurring between people with molecular linkage (Figure 2). There were a total of 6768 links between nodes at d = 1.5%. Transmission among MMSC (3779/6768 = 55.8%) and between MMSC and HET (2046/6768 = 30.2%) were the predominant routes of transmission in clusters of ART-naive individuals, followed by transmission among HET (448/6768 = 6.7%) and among PWID (206/6768 = 3.0%), between PWID and MMSC or HET (479/6768 = 7.1% combined), and among MMSC-PWID and four other risk groups (64/6768 = 0.9%). Mixtures of potential transmission pair types were evident in all subtype cluster sizes, including MMSC–MMSC, MMSC–HET, and HET–HET in the largest subtype B and F1 clusters. In contrast, the CRF01_AE clusters had more potential transmission among PWID and other risk groups. Some clusters containing all or mostly male patients also had both MMSC–MMSC and MMSC–HET transmission pairs. Analysis of the gender of the HET individuals in these MMSC–HET transmission links identified most (96.2%, 1968/2046) as male, suggesting unreported bisexual behavior. Our results do not infer directionality but rather describe the potential type of transmission occurring in the clusters.

Our findings indicate a mixture of more recent and prevalent infections in these highly genetically diverse transmission clusters. We also found that the majority of men reporting HET transmission risk (271/1053) have genetic links to MMSC that are suggestive of undisclosed MMSC behavior.

#### 3.4.2. Clustering of HIV-1 Sequences with SDRMs

Transmission network analysis at d = 1.5% identified a 14-member cluster containing 13 HIV-1 subtype B sequences with the L90M PI and one with the T215TS NRTI SDRMs (Figure 2). The majority of the genetic links in this cluster were between MMSC (40/58 = 69%), followed by MMSC–HET (17/58 = 29.3%), and one HET–HET link. Five subtype A1 sequences with the K103N NNRTI clustered with an A1 sequence that had no DRMs. All 15 genetic links in this six-member cluster were between MMSC. Three CRF12_BF sequences with K103N NNRTI DRMs clustered with a CRF12_BF sequence with K103N, V106VI and E138EK NNRTI DRMS. All six genetic links in this four-member cluster were between MMSC. Two subtype F1 sequences from a HET and MMSC with the M46L PI and the E138A NNRTI DRMs clustered together, while two additional subtype F1 sequences from a HET and MMSC had combinations of NRTI (L74V, Y115F, M184V and M41ML, respectively) and NNRTI (V106VU, E138G and V106I, respectively) DRMs but clustered with 57 other F1 sequences without DRMs. Seven sequences (five subtype B, one CRF01_AE, one URF F1B) with either a PI, NRTI, or an NNRTI DRM were present in clusters of sizes seven to 100, including an F1B URF with the T215I DRM that clustered with 24 F1B sequences without DRMs. The remaining 26 sequences with SDRMs (3 with PI, 17 with NRTI, 12 with NNRTI, 1 with PI + NRTI, and 3 with NRTI + NNRTI) did not cluster and consisted of subtype B (n = 13), CRF02_AG (n = 4), CRF01_AE (n = 2), and one each of subtypes A1D, A1CD, A6, CRF12_BF, BF1, CRF39_cpx, and URF.

We evaluated clusters with and without SDRMs as well as non-clustering sequences with and without SDRMs to check for association with gender, transmission risk, and subtype (Table 3). Our clustering results show continued onward spread of some SDRMs in transmission networks consisting mostly of MMSC and that clustering of SDRMS was associated with subtype.

## 4. Discussion

In this nationally representative study, we analyzed transmitted drug resistance, HIV-1 diversity, and transmission clusters by combining demographic, molecular and virological data from 1053 treatment-naive HIV-1 individuals diagnosed in Bulgaria during 2012–2020. We found that the overall rate of SDRMs in Bulgaria remains low (5.7%), occurring against a background of very high genetic diversity, and was similar to the 5.2% rate found in our previous study for the period 1988 to 2011. However, in our current study we observed a shift from a higher prevalence of heterosexual and PWID infections during 1988–2011 to MMSC, which is also consistent with MMSC now having the highest HIV infection rate in Bulgaria [17]. The prevalence of SDRMs in Bulgaria is lower than rates found in most neighboring countries, with the highest rate of 16% reported recently for Croatia [11]. The last representative data from the SPREAD (Strategy to Control SPREAD of HIV Drug Resistance) program of the European Society for Translational Antiviral Research (ESAR) showed that, from 2008–2010, the overall TDR prevalence in European countries was 8.3% [9]. A more recent study showed that, during 2011–2019, the TDR prevalence in Europe increased to 12.8% [36].

Similarly to the results of the 2016 pan-European study conducted by Hofstra et al., SDRMs in Bulgaria were most common in MMSC (60%) and less common among HET (36.6%) and PWID (3.3%). Our transmission network analysis of sequences from ART-naive people showed that most phylogenetic clusters in Bulgaria were composed mostly of MMSC, with the potential to facilitate the accelerated spread of resistance mutations among these individuals. Indeed, we identified a 14-member cluster of subtype B sequences from 12 MMSC and two male HET with 13 PI and one NRTI SDRMs, demonstrating transmission among these individuals, with opportunities for further spread among HET. While most HET in these MMSC clusters are likely to be non-disclosed MMSC, we have previously shown spread among female HET and MMSC [22,23]. We also identified two smaller MMSC clusters consisting of either A1 or CRF12_BF subtypes that contained NNRTI SRDMs. Nonetheless, other large clusters had a single introduction of a SDRM that did not spread and almost 43% were singletons, which likely contributed to the observed low SDRM rate in Bulgaria and the absence of an association of SDRMs with clustering. We also observed fluctuating SDRM rates across transmission categories over time, with a high prevalence in HET at the start of the study (2012–2015) followed by a decline, with a corresponding increase in SDRMs in MMSC in subsequent years (2015–2019). Although this trend was not significant because of the sparseness of the data each year, there are some likely explanations for these observations. The initial higher rate among HET was probably from the early introduction of HIV into this population with the longest history of ART [21]. In contrast, HIV was introduced much later (around 2010) among MMSC in Bulgaria but has now become the predominant transmission route, likely contributing to SDRM transmission in this group [22].

ART in Bulgaria follows EACS guidelines and is currently being upgraded to the latest version (V11.1); however, most SDRMs identified here can impact HIV treatment [37]. The most common NRTIs identified in our current study were the M184V and thymidine analog mutation M41L, in eight patients each. M184V is frequently targeted by 3TC/FTC, which have been the most widely used drugs for many years, both globally and in Bulgaria [38]. M184V can reduce susceptibility to 3TC and FTC > 200-fold and has also been associated with PrEP failure when present in drug-naive infections such as our study population [39]. Nonetheless, other studies have shown that M184V can be beneficial in increasing the susceptibility of HIV-1 to tenofovir [40]. M41L continues to spread, but is likely a relic of previously discontinued treatment with AZT. Likewise, the SDRMs T215C/D/S are revertants of the primary AZT resistance mutations T215Y/F that continue to spread in drug-naive people [38]. The common nonpolymorphic NRTI resistance mutation V75M, identified in people receiving d4T/3TC-containing regimens and occurring predominantly in CRF01_AE infection, was found in one patient with this subtype [38]. Since 2009, the WHO has advised against using d4T due to its long-term, irreversible side effects. Nonetheless, studies have shown slow declines in d4T usage in Eastern Europe, including Bulgaria, which may explain our findings [41]. A rare mutation, T69D, was identified in a Bulgarian male who reported Belgium as the potential country in which his infection was acquired. T69D is a nonpolymorphic NRTI-selected mutation that reduces susceptibility to ddI and possibly d4T [42].

The most frequent NNRTI was K103N, identified in 15 patients, which is the most commonly transmitted nonpolymorphic NNRTI-drug-resistance mutation seen in people receiving NVP and EFV, drugs that were in use in Bulgaria during the study period [43,44]. Nine people with the K103N were in two different clusters of MMSC, suggesting the DRM is spreading in Bulgaria among individuals with this type of risk behavior. The presence of K103N before starting the WHO-recommended first-line regimen tenofovir/lamivudine/efavirenz is associated with an increased risk of virological failure [45]. However, longitudinal viral load results were not available for these individuals to facilitate the evaluation of this possibility. A related DRM, K101E, was found in three patients with subtype B infection, one female HET and two MMSC, and all were from different regions of the country, suggesting this DRM may be widespread across Bulgaria. K101E is a nonpolymorphic mutation commonly found in people receiving each of the NNRTIs except DOR [41]. 

Twenty-two people had SDRMs to PIs, of which L90M was the most common, followed by M46I/L. All L90M carriers were infected with subtype B, are members of an MMSC–HET transmission cluster and were from different regions in Bulgaria, and three of them reported probable infection abroad. Phylogenetic analysis following removal of the L90M resistance sites showed clustering identical to that identified using MicrobeTrace, demonstrating that clustering was not affected by this mutation. L90M is selected primarily by SQV, NFV, IDV, and ATV, while M46L is usually selected by all PIs except SQV and DRV [46]. SQV and other NNRTI drugs were commonly used globally, including in Bulgaria, which probably explains the prevalence of NNRTI-associated mutations among our study participants; however, NNRTIs are currently not often used in modern therapeutic regimens [46,47,48,49].

SDRMs were identified in 5.9% of people reporting infection acquired abroad, sequences from some of which clustered, demonstrating that SDRMs likely continue to be imported into Bulgaria with some onward spread. Analysis of the geographical distribution of individuals with SDRMs showed that half of the cases were found in the capital city of Sofia, where the largest number of HIV-1 infections in our study were registered. However, SDRMs were also identified in people from 15 other regions of Bulgaria, including remote locations, demonstrating that SDRMs are widespread in the country, despite a low overall prevalence.

Our sequence analyses confirmed the high HIV-1 diversity in Bulgaria observed in our previous reports, showing high diversity in both individuals infected within Bulgaria and those moving to or working in Bulgaria from other countries [16,17,19,20,21,22,23]. The majority of infections (60%) were classified as subtype B, which is comparable to that (65.8%) seen in most European countries and in those adjacent to Bulgaria [9], except for Romania and Albania, where the most prevalent subtypes are F1 (80.3%) and A1 (56.1%) [50,51]. Interestingly, eight subtype B infections with SDRM were seen in people who acquired their infections abroad, including four who were in a 14-member cluster, indicating that these SDRMs are being introduced into Bulgaria from other countries and spreading in Bulgaria.

CRF01_AE was the second most prevalent subtype observed in our study population, and this has increased significantly in Bulgaria, likely due to a local outbreak among PWIDs in the capital, Sofia, in 2009–2011. Bulgaria now has the highest prevalence of CRF01_AE among the Balkan countries [20,23,50]. We also identified 23 additional distinct subtypes, including recombinant, complex and unique recombinant forms, including three subtypes that are predominant in neighboring countries and which were likely imported into Bulgaria. For example, subtype F1 is dominant in Romania, sub-subtype A6 is prevalent in Russia and Ukraine, and subtype A1 is predominant in Albania [50,51,52,53]. Subtypes F1 and A1 appear to have spread quickly after being introduced into Bulgaria, with the formation of a large 59-member cluster of the 73 total F1 sequences and the formation of an 18-member, 6-member, and triad of the total 39 A1 sequences. In contrast, sub-subtype A6 has had a more limited person-to-person spread in Bulgaria, with only two small clusters identified (a dyad and triad) of the 33 total A6 sequences. These results also suggest that A6 has been introduced more frequently than F1 and A1, given that 28 were singletons, compared to 10 for F1 (four were members of dyads) and 12 for A1.

Subtypes were not evenly distributed among individuals with various transmission risk behaviors. Most subtype B sequences were found in men, of whom the majority were MMSC. These results are consistent with the characteristics of the epidemic in Western Europe and those from our previous study [21,22,54]. In contrast, CRF01_AE was found mostly in PWIDs and rarely in MMSCs. The differences in the prevalence of these subtypes among MMSC and PWIDs are most likely a consequence of their initial introduction into these groups, followed by rapid dissemination among these populations, as described in our previous studies [20,21,22,55,56]. This result is supported by our finding of transmission clusters containing sequences from people with different transmission risk behaviors.

Our findings have some limitations, including exclusion of some people from the study because they did not meet the participant selection criteria. For example, some were rejected because they had previously received therapy in another country. Only 52.4% of the 2010 individuals diagnosed with HIV-1 during 2012–2020 met the study criteria and participated. Although we expect that a cross-sectional study design will provide a representation of the factors that may be involved in TDR in Bulgaria, it is possible that this subset of eligible individuals may not be truly representative of all reported cases. The effect of this possible sampling bias in our analyses may also influence our reporting of TDRMs rates, subtype diversity and the distribution of both in different population groups. Additionally, self-reporting of epidemiological data used in this study could introduce recall biases, especially for those reporting infections acquired abroad, which could affect the reported subtype prevalence by country of origin or potential HIV transmission route.

Finally, our results will supplement those from other studies to expand and update the criteria for the WHO SDRM list that was adopted and validated over 12 years ago, because new antiretroviral therapies have been implemented since then [44]. A more regular review and modification of the WHO SDRM list is warranted. Furthermore, the 2009 SDRM list does not include the complete list of mutations that are detected with standard resistance analysis for the monitoring of patients on therapy, which should be considered in an updated version [44].

## 5. Conclusions

Our study provides valuable information on the transmission dynamics of HIV drug resistance in the context of high genetic diversity. We found a low and relatively stable SDRM prevalence amid high subtype diversity in treatment-naive, HIV-1-infected individuals in Bulgaria. These mutations, with the exception of M184V, are associated with low costs to replicative fitness and therefore are able to persist for several years following earlier treatments. While these stable SDRMs results are encouraging, we found that over half of the sequences with SDRMs were found in transmission clusters containing MMSC, and this may be indicative of underreporting of MMSC status. While there appears to be onward spread of SRDMs in Bulgaria, it may not have clinical consequences for antiretroviral regimens in use today. The contribution of transmission clusters to the spread of SDRMs within and across specific groups, with increased chances for infection and introduction and spread of diverse HIV subtypes into Bulgaria, remains a public health concern. Our study demonstrates the importance of focused surveillance to further track SDRMs and transmission clusters in Bulgaria, in order to facilitate enhanced HIV prevention efforts.

## Figures and Tables

**Figure 1 viruses-15-00941-f001:**
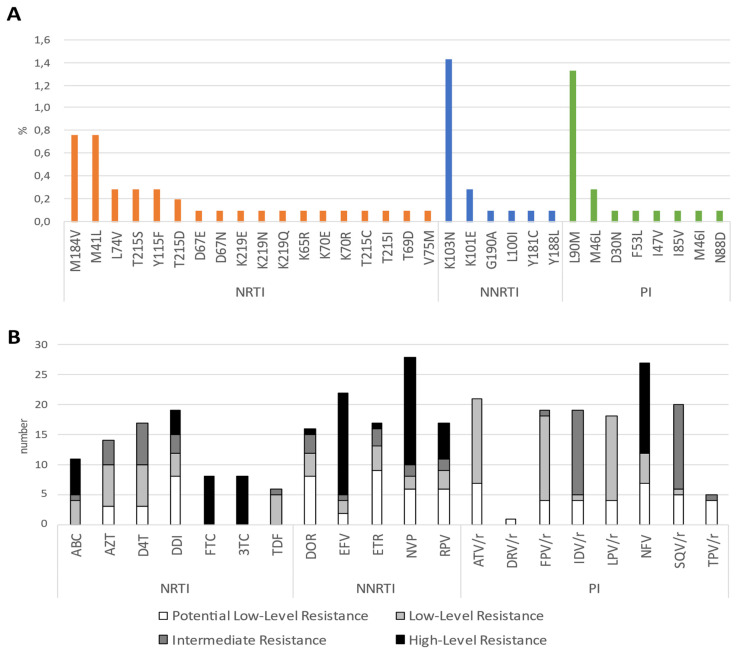
Transmitted drug resistance in Bulgaria and predicted phenotypic resistance. (**A**) Proportion of specific drug resistance mutations in the HIV-1 polymerase sequences of newly diagnosed, antiretroviral-naive patients and (**B**) Number of patients with predicted phenotypic resistance based on Stanford scores to specific antiretroviral drugs. NRTI, nucleoside reverse transcriptase inhibitor; NNRTI, non-nucleoside reverse transcriptase inhibitor; PI, protease inhibitor. ABC = abacavir, AZT = zidovudine, D4T = stavudine, DDI = didanosine, FTC = emtricitabine, 3TC = lamivudine, TDF = tenofovir disoproxil fumarate, DOR = doravirine, EFV = efavirenz, ETR = etravirine, NVP = nevirapine, RPV = rilpivirine, ATV/r = ritonavir-boosted atazanavir, DRV/r = ritonavir-boosted darunavir, FPV/r = ritonavir-boosted fosamprenavir, IDV/r = ritonavir-boosted indinavir, LPV/r = ritonavir-boosted lopinavir, NFV = nelfinavir, SQV/r = ritonavir-boosted saquinavir, TPV/r = ritonavir-boosted tipranavir.

**Figure 2 viruses-15-00941-f002:**
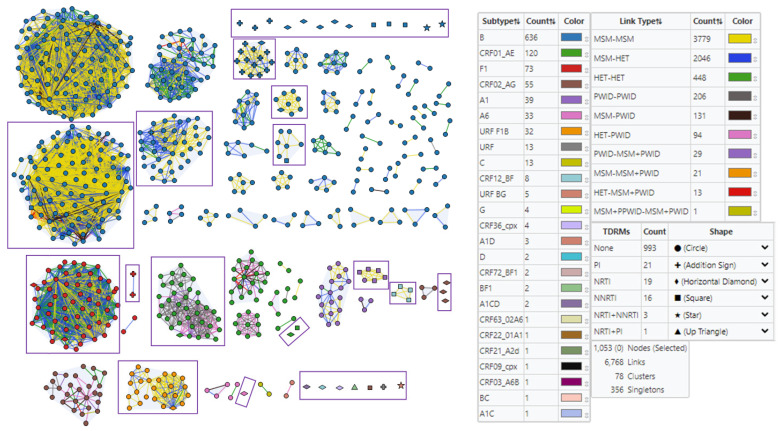
Identification and characterization of HIV transmission networks in antiretroviral-naive Bulgarians using MicrobeTrace. HIV-1 polymerase (*pol*) sequence clusters identified using a genetic distance cutoff (d) of 1.5%. Nodes are colored by final subtype classification using a combination of sequence analysis methods. Node shape indicates type of resistance mutation: PI, protease inhibitor; NRTI, nucleoside reverse transcriptase inhibitor; NNRTI, non-nucleoside reverse transcriptase inhibitor. Clusters are grouped by size and subtype, and singletons with surveillance drug resistance mutations (SDRMs) are also shown, whereas those non-clustering sequences without resistance mutations were excluded. Clusters and singletons with SDRMs are shown in boxes. Edges (links) are colored by transmission risk category (MMSC, male-to-male sexual contact; HET, heterosexual transmission; PWID, people who inject drugs; MMSC + PWID, MMSC who also report injecting drugs).

**Table 1 viruses-15-00941-t001:** Prevalence and characteristics of surveillance drug resistance mutations (SDRMs) in antiretroviral treatment-naive patients in Bulgaria.

Characteristic	HIV-1 Infections with SDRMs (n = 60)	HIV-1 Infections without SDRMs (n = 993)	*p* ^1^
**Gender**			**0.446**
Male	54 (90.0%)	851 (85.7%)	
Female	6 (10.0%)	142 (14.3%)	
**Likely route of HIV infection** ^2^			**0.242**
HET	22 (36.6%)	379 (38.2%)	
MMSC	36 (60.0%)	498 (50.2%)	
PWID	2 (3.3%)	99 (10.0%)	
MMSC + PWID	0 (0.0%)	17 (1.7%)	
**Likely country of infection**			**0.596**
Bulgaria	48 (80.0%)	824 (82.9%)	
Abroad	12 (20.0%)	169 (17.0%)	
**Migrants** ^3^			**0.239**
Yes	15 (25.0%)	187(18.8%)	
No	45 (75.0%)	806 (81.2%)	
**Individuals incarcerated**			**0.717**
Yes	1 (1.7%)	34 (3.4%)	
No	59 (98.3%)	959 (96.6%)	
**Blood donors**			**0.043**
Yes	0 (0.0%)	56 (5.6%)	
No	60 (100.0%)	937 (94.4%)	
**Sex workers**			**0.537**
Yes	2 (3.3%)	29 (2.9%)	
No	58 (96.7%)	964 (97.1%)	
**Pregnant Women**			**0.643**
Yes	2 (3.3%)	22 (2.4%)	
No	58 (96.7%)	971 (97.8%)	
**STI** ^4^			**0.926**
Yes	15 (25.0%)	243 (24.5%)	
No	45 (75.0%)	750 (75.5%)	
**Subtype** ^5^			**<0.0001**
B	32 (53.3%)	604 (60.8%)	
F1	4 (6.7%)	69 (6.9%)	
CRF02_AG	4 (6.7%)	51 (5.1%)	
A1	5 (8.3%)	34 (3.4%)	
CRF12_BF	5 (8.3%)	3 (0.3%)	
Other	10 (16.7%)	232 (23.4%)	

^1^ *p* < 0.05 was considered statistically significant. ^2^ HET, heterosexual transmission; MMSC, male-to-male sexual contact; PWID, person who injects drugs; MMSC + PWID, MMSC also reporting PWID behavior. ^3^ Migrants were defined as Bulgarians likely infected while living abroad and foreigners who were diagnosed while working in Bulgaria. ^4^ STI, sexually transmitted infection other than HIV, including *Chlamydia trachomatis*, hepatitis B, *Trichomonas vaginalis*, syphilis, and genital herpes. ^5^ Subtypes present in <4 people in total were collapsed into “Other” for the statistical analysis.

**Table 2 viruses-15-00941-t002:** High HIV-1 genetic diversity in antiretroviral-naive Bulgarians.

HIV-1 Subtype	HIV-1 Subtype (%)	Women (%)	Men (%)	Bulgarians Infected in Bulgaria (%)	Bulgarians Infected Abroad (%)	Foreigners Diagnosed in Bulgaria (%)	HET ^1^ (%)	MMSC ^2^ (%)	PWID ^3^ (%)	MMSC + PWID ^4^ (%)	Any SDRMs (%)
A1	39 (3.7)	3 (2)	36 (4.0)	32 (3.8)	5 (3.3)	2 (4)	9 (2.2)	28 (5.2)	2 (2)		5 (8.3)
A1C	1 (0.1)	1 (0.7)		1 (0.1)			1 (0.2)				
A1CD	2 (0.2)	1 (0.7)	1 (0.1)	1 (0.1)	1 (0.7)		2 (0.5)				1 (1.7)
A1D	3 (0.3)	2 (1.4)	1 (0.1)	1 (0.1)		2 (4)	2 (0.5)	1 (0.2)			1 (1.7)
A6	33 (3.1)	12 (8.1)	21 (2.3)	19 (2.2)	7 (4.6)	7 (14)	24 (6)	8 (1.5)	1 (1)		1 (1.7)
B	636 (60.4)	49 (33.1)	587 (64.9)	533 (62.6)	88 (57.9)	15 (30)	213 (53.1)	404 (75.7)	15 (14.9)	4 (23.5)	32 (53.3)
BC	1 (0.1)		1 (0.1)		1 (0.7)			1 (0.2)			
BF1	2 (0.2)		2 (0.2)	1 (0.1)	1 (0.7)		2 (0.5)				1 (1.7)
C	13 (1.2)	9 (6.1)	4 (0.4)	8 (0.9)	3 (2)	2 (4)	12 (3)			1 (5.9)	
CRF01_AE	120 (11.4)	32 (21.6)	88 (9.7)	101 (11.9)	17 (11.2)	2 (4)	40 (10)	10 (1.9)	64 (63.4)	6 (35.3)	3 (5)
CRF02_AG	55 (5.2)	17 (11.5)	38 (4.2)	43 (5.1)	6 (3.9)	6 (12)	30 (7.5)	6 (1.1)	13 (12.9)	6 (35.3)	4 (6.7)
CRF03_A6B	1 (0.1)		1 (0.1)	1 (0.1)					1 (1)		
CRF09_cpx	1 (0.1)		1 (0.1)			1 (2)	1 (0.2)				
CRF12_BF	8 (0.8)	1 (0.7)	7 (0.8)	6 (0.7)	1 (0.7)	1 (2)	2 (0.5)	6 (1.1)			5 (8.3)
CRF21_A2D	1 (0.1)		1 (0.1)	1 (0.1)					1 (1)		
CRF22_01A1	1 (0.1)	1 (0.7)				1 (2)	1 (0.2)				
CRF36_cpx	4 (0.4)	1 (0.7)	3 (0.3)	3 (0.4)		1 (2)	3 (0.7)		1 (1)		1 (1.7)
CRF63_02A6	1 (0.1)		1 (0.1)	1 (0.1)			1 (0.2)				
CRF72_BF1	2 (0.2)		2 (0.2)	2 (0.2)			2 (0.5)				
D	2 (0.2)	1 (0.7)	1 (0.1)			2 (4)	2 (0.5)				
F1	73 (6.9)	12 (8.1)	61 (6.7)	60 (7.1)	10 (6.6)	3 (6)	37 (9.2)	35 (6.6)	1 (1)		4 (6.7)
G	4 (0.4)	2 (1.4)	2 (0.2)		2 (1.3)	2 (4)	3 (0.7)	1 (0.2)			
URFs ^5^	50 (4.7)	4 (2.7)	46 (5.1)	37 (4.3)	10 (6.6)	3 (6)	14 (3.5)	34 (6.4)	2 (2)		2 (3.3)
**Total**	**1053**	**148 (14.1)**	**905 (85.9)**	**851 (80.8)**	**152 (14.4)**	**50 (4.7)**	**401 (38.1)**	**534 (50.7)**	**101 (9.6)**	**17 (1.6)**	**60 (5.7)**

^1^ HET, heterosexual transmission. ^2^ MMSC, male-to-male sexual contact. ^3^ PWID, person who injects drugs. ^4^ MMSC + PWID, MMSC also reporting as a PWID. ^5^ URFs, unique recombinant forms including F1B (n = 32), BG (n = 5), and 13 URFs that were not further characterized.

**Table 3 viruses-15-00941-t003:** Comparative characteristics of people with HIV-1 surveillance drug resistance mutations (SDRMs) with sequences in clusters versus those that do not cluster.

Characteristic ^1^	Sequences with SDRMs in Clusters (n = 34)	Sequences without SDRMs in Clusters (n = 663)	*p* Value ^2^	Non-Clustering Sequences with SDRMs (n = 26)	Non-Clustering Sequences without SDRMS (n = 330)	*p* Value ^3^
**Gender**			0.104			1.0
Male	34 (100%)	**604 (91.1%)**		20 (76.9%)	247 (75.9%)	
Female	0 (0%)	**59 (8.9%)**		6 (23.1%)	83 (25.1%)	
**Likely route of HIV infection**			0.442			0.846
HET	6 (17.6%)	**190 (28.7%)**		17 (65.4%)	189 (57.3%)	
MMSC	28 (82.4%)	**384 (57.9%)**		8 (30.8%)	114 (34.5%)	
PWID	0 (0%)	**80 (12.1%)**		2 (7.7%)	19 (5.8%)	
MMSC + PWID	0 (0%)	**9 (1.4%)**		0 (0%)	8 (2.4%)	
**Subtype**			<0.0001			0.250
B	19 (55.9%)	**444 (83.7%)**		13 (50.0%)	160 (48.5%)	
F1	4 (11.8%)	**59 (8.9%)**		0 (0%)	10 (3.0%)	
CRF02_AG	0 (0%)	**30 (4.5%)**		4 (15.4%)	21 (6.4%)	
A1	5 (14.7%)	**22 (3.3%)**		0 (0%)	12 (3.6%)	
CRF12_BF	4 (11.8%)	**0 (0%)**		1 (3.8%)	3 (0.9%)	
Other	2 (5.9%)	**108 (16.3%)**		8 (30.8%)	124 (37.6%)	

^1^ HET, heterosexual transmission; MMSC, male-to-male sexual contact; PWID, people who inject drugs; MMSC + PWID, MMSC who also report injecting drugs. ^2^ *p* value for comparing the sequences with SDRMs in clusters to the sequences without SDRMs in clusters. ^3^ *p* value for comparing the non-clustering sequences with SDRMs to the non-clustering sequences without SDRMS.

## Data Availability

A random and anonymized subset of 10% of the HIV *pol* sequences generated in our study have been deposited at GenBank, following the guidance of other groups for the privacy and confidentiality protections of HIV-infected patients [57]. Additional data presented in this study are available upon request from the corresponding author.

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
