# Peer review of "Transmitted HIV Drug Resistance in Bulgaria Occurs in Clusters of Individuals from Different Transmission Groups and Various Subtypes (2012–2020)"

_viruses, 2023, doi:10.3390/v15040941_

Round 1
Reviewer 1 Report
In this article, authors characterized transmitted drug resistance in Bulgaria from 1052 polymerase sequences from antiretroviral therapy-naïve individuals between 2012-2020. The SDRM rate was low (5.7%) and included resistance mutations associated with NRTIs, NNRTIs and protease inhibitors. The authors found that a majority of the SDRMs were present in transmission clusters involving male-to-male sexual contact. The study provides a valuable picture of transmission dynamics of HIV drug resistance in Bulgaria that can inform strategies to suppress transmission.
General critique. The study provides a thorough analysis of transmission of drug resistance mutations in Bulgaria. The overall conclusion that transmission among MMSCs is more prevalent than heterosexual contacts seems solid. If the authors can provide a statistical comparison of the transmission links described from lines 252-264, that would further strengthen the conclusions.
Specific comments. Overall, many of the points seem to be repeated throughout the manuscript, especially in the Discussion section, which could be considerably shortened by removing points that merely restate the points already made.
Figure 1 shows fluctuations in transmitted drug resistance over an 8-year period between 2012 to 2020. It is not clear whether these fluctuations represent statistically significant differences or stochastic changes. Can the authors provide some statistical analysis to determine whether the apparent increase in transmitted drug resistance in the MMSC group is significant at year 2017, compared to the low point in 2015?
Author Response
Viruses March 30, 2023
Dr. Eric O. Freed
Editor-in-Chief
MDPI
St. Alban-Anlage 66,
4052 Basel, Switzerland
Tel. +41 61 683 77 34
Fax: +41 61 302 89 18
Dear Dr. Freed,
Thank you for considering our manuscript entitled "Transmitted HIV drug resistance in Bulgaria is mainly identified in clusters of persons from different transmission groups and various subtypes (2012-2020)" with manuscript ID: viruses-2247440 for publication in Viruses. We have revised the paper carefully in the manner suggested by the reviewers. Please find attached our point-by-point response to the comments of the reviewers. We hope you find that the modified manuscript is now suitable for publication.
We look forward to hearing from you again.
Sincerely,
Ivailo Alexiev, Ph.D.
National Reference Laboratory of HIV,
National Center of Infectious and Parasitic Diseases
44a Gen. N. Stoletov Blvd
1233 Sofia, Bulgaria
tel & fax # 359 2 9318071
Email: ivoalexiev@yahoo.com
Response to reviewers:
Reviewer 1
Comments and Suggestions for Authors
In this article, authors characterized transmitted drug resistance in Bulgaria from 1052 polymerase sequences from antiretroviral therapy-naïve individuals between 2012-2020. The SDRM rate was low (5.7%) and included resistance mutations associated with NRTIs, NNRTIs and protease inhibitors. The authors found that a majority of the SDRMs were present in transmission clusters involving male-to-male sexual contact. The study provides a valuable picture of transmission dynamics of HIV drug resistance in Bulgaria that can inform strategies to suppress transmission.
General critique. The study provides a thorough analysis of transmission of drug resistance mutations in Bulgaria. The overall conclusion that transmission among MMSCs is more prevalent than heterosexual contacts seems solid. If the authors can provide a statistical comparison of the transmission links described from lines 252-264, that would further strengthen the conclusions.
We thank the reviewer for this suggestion. To test the validity of the hypotheses, we conducted a comprehensive statistical analysis, which is now described in the new section 3.4. Association of selected demographic characteristics in persons with SDRMs. In addition, we have added a new Table 3.
Specific comments. Overall, many of the points seem to be repeated throughout the manuscript, especially in the Discussion section, which could be considerably shortened by removing points that merely restate the points already made.
We thank the reviewer for this suggestion. We have now amended and shortened the discussion section as suggested.
Figure 1 shows fluctuations in transmitted drug resistance over an 8-year period between 2012 to 2020. It is not clear whether these fluctuations represent statistically significant differences or stochastic changes. Can the authors provide some statistical analysis to determine whether the apparent increase in transmitted drug resistance in the MMSC group is significant at year 2017, compared to the low point in 2015?
We thank the reviewer for this suggestion. We statistically analyzed the SDRMs by year but found these were not significant likely due to the sparseness of the data for each year. We have now removed Figure 1 but describe the results and provide possible explanations for them.
Submission Date
12 February 2023
Date of this review
09 Mar 2023 19:58:34
Reviewer 2
Comments and Suggestions for Authors
Alexiev et al. here describe the results of baseline drug resistance analyses on patients newly diagnosed between 2012 and 2020 in Bulgaria. This manuscript is well written but I have some remarks/recommendations:
- It would be of added value if some additional information could be given on the overall HIV prevalence in Bulgaria and how this compares with the neighboring countries? It is mentioned that 2,010 individuals were diagnosed with HIV-1 during 2012-2020. This seems to be a very low prevalence.
We thank the reviewer for this comment. We already provided HIV prevelances in neighboring countries in the introduction but now add additional details of the outbreak in Bulgaria with references as suggested.
- Additional information on the current treatment guidelines in the country may also be of interest. This would allow the reader to make a better estimate on the impact of particular mutations.
We thank the reviewer for this comment. We have added additional information to the Discussion section along with a reference. “ART in Bulgaria follows EACS Guidelines and is currently being upgraded to the latest version (V11.1), however, most SDRMs we identified here can impact HIV treatment.
- The algorithm that is used for identification of transmitted drug resistance dates from more than 10 years ago. It is essential to reflect a bit on the potential consequences thereof: scoring of mutations that impact drugs no longer in use (already briefly discussed) but also, and more importantly, the possibility of missing mutations that impact newer drugs.
Thanks for this suggestion. We have expanded this point in the discussion to reflect the concern that the SRDM list has not been updated in some time and that it is imperative for clinicians and public health experts that this be done more regularly.
- In the paragraph between lines 353 and 359 on page 7 the clustered occurrence of the L90M mutation is discussed. Maybe I missed something, but I find it very difficult to understand how, in a cluster of 13 members with the L90M mutations, three cluster members are supposed to have attracted their infection abroad. Except when all 3 have visited the same place, the chance that they have been infected with a related virus seems almost impossible. Do these sequences also cluster if codon 90 is removed?
We thank the reviewer for this comment. We have changed the text in this section to clarify the finding and indicate the results were not affected by removal of the L90M mutation.
- Please add the total length of the sequences used for phylogenetic analysis.
We have now added in the Sequence analyses section ( 2.2) the length of the aligned sequences used for the phylogenetic analyses.
Submission Date
12 February 2023
Date of this review
01 Mar 2023 11:14:01
Reviewer 3
Comments and Suggestions for Authors
The authors presented data on drug resistance among patients without ARV experience in Bulgaria for the period 2012-2020. This is important work as drug resistance is currently one of the major barriers to achieving the 95-95-95 targets. This is also a very informative work, since the sample density is very high (more than 50%) and the analyzed period is quite long.
At the same time, according to the reviewer, the authors were not very good at presenting their results. The notes listed below describe the desired changes to the manuscript.
- It would be nice to see more statistics data in the Introduction section. In particular, it would be useful to include information on the total number of people living with HIV in Bulgaria, the incidence trend over the study period. It is also important to provide information on standards for recommended first-line ART regimens for 2012-2020.
We thank the reviewer for this comment. We have now include additional epidemiological information regarding the outbreak of HIV in Bulgaria in the introduction and the Discussion sections along with a reference.
- In a number of places in manuscript, data on the prevalence of HIVDR are poorly presented and confuse the reader. For example, in section 3.3 (lines 193-194) it is indicated that the prevalence of SDRM is highest among MSM (60%), then among heterosexuals (36.6%) and further among IDUs (3.3%). However, this information is of little use. It would be clearer and more informative here and throughout the text to indicate the prevalence of DR, taking into account the number of people in the cohort, namely MSM - 6.7% (36/534), HETERO - 5.5% (22/401), IDUs - 2.0% (2/101) and add confidence interval. Another example is on lines 205-209. Indeed, the greatest number of mutations was found in subtype B viruses (32/60). However, the DR prevalence in this subtype is below average at 5.0% (32/636). It is much more important that the prevalence of mutations, for example, in CRF12_BF is 62.5% (5/8), and in subtype A1 12.8% (5/39).
We thank the reviewer for this comment. We conducted a comprehensive statistical analysis, which is now described in the new section 3.4. Association of selected demographic characteristics in persons with SDRMs. In addition, we have added a new Table 3.
- How can one explain the cyclical growth of the DR, shown in Figure 1? Are these differences in prevalence statistically significant? The manuscript does not indicate the distribution of the analyzed clinical samples by year, so it is not possible to estimate the breadth of the confidence interval. It would be useful to indicate such a confidence interval on the graph (Figure 1) for each annual value of HIVDR prevalence.
We thank the reviewer for this suggestion. We statistically analyzed the SDRMs by year but found these were not significant likely due to the sparseness of the data for each year. We have now removed Figure 1 from the manuscript.
- The Y scale in Figure 2B is probably wrong. Stanford scores are not expressed as a percentage.
We thank the reviewer for this comment. Figure 2 (B), now 1 (B), presents the number of patients with predicted phenotypic resistance using Stanford scores to specific antiretroviral drugs. This clarification has been added to the text of the figure.
- The conclusion in section 3.3 (lines 234-235) should be corrected and softened. Indeed, the MSM group had the highest number of mutations, but the group size was also the largest. If we calculate the prevalence of mutations in different groups, then it turns out that the difference between the prevalence in the MSM and HETERO groups is not statistically significant.
We thank the reviewer for this comment. We have now updated the text as suggested.
- Section 3.4.2 is difficult to understand. It would be useful to add a table in which the first column indicated the analyzed parameter (sex, transmission path, subtype, etc.), the second column indicated the number (and %) of clustered sequences, and the third column indicated the number (and %) of clustered sequences with DR mutations, in the fourth - the number (and %) of non-clustered sequences with DR, in the fifth - the significance (p) of differences between the second and third columns. So, it will be visually represented how often sequences with DR are clustered.
Thanks for this great suggestion. We have added Table 3 per your suggestion and conducted the statical analyses of SDRMs in clusters and not in clusters with the variables suggested.
- Lines 362-364. Bad wording. It appears that DR mutations were found in one fifth of PLHIV infected abroad. The reason is the same as above. When talking about a specific cohort, in this case people infected outside the country, it is more correct to discuss the prevalence of mutations in a specific cohort of patients.
We thank the reviewer for noticing this. We have amended the text and added the correct values.
- Lines 367-370. Bad wording. The reason is the same.
We thank the reviewer for noticing this and we have removed that sentence from the manuscript.
Reviewer 4
Comments and Suggestions for Authors
This study evaluated the prevalence of circulating HIV-1 drug resistance mutations in Bulgaria from 2012-2020 and compared their findings to previous data generated from 1988-2011. They used a surveillance approach to determine the prevalence of drug resistance mutations (DRMs). The study is significant because the prevalence of DRMS can be as high as 16% in other regions and it is serious enough that drug resistance testing is recommended before HIV+ individuals begin ART.
The authors evaluated HIV-1 pol and protease sequences in 1,053 ART-naïve participants, which was a subset of a larger cohort of 2,010 ART naïve individuals diagnosed with HIV-1 in Bulgaria. Of the 1,053 participants, 60 were found to harbor DRMs, which is 5.7% and is similar to the previous time frame. They also found that circulating HIV-1 is diverse. The majority subtype was B (60%) but they also observed F1, CRF02_AG, A1, CRF12_BF, etc. They found that most DRMs were present in transmission clusters of men who have sexual contact with men. They also reported that while the prevalence is similar to the previous period, there appears to be a shift in the route of dissemination with men who have sex with men now being higher than heterosexual contact or injecting drug use. Overall this is an interesting and informative study; however, there are major concerns with the statistical analyses in which the authors evaluated factors that could be associated with harboring DRMs, and other issues as detailed below.
The statistical comparisons are flawed and should be revised. Table 1 should be SDRM+ vs. SDRM-. The comparison of SDRM+ to the total (SDRM+ and SDRM-) is not rational. All text and analyses related to Table 1 should be revised with an appropriate statistical method.
We thank the reviewer for this suggestion. We agree that using the totals in Table 1 can be confusing. We have now changed Table 1 to compare HIV-1 infections with SDRMs to those without SDRMs with statistical analyses and have also updated the text accordingly. Overall, the results have not changed. In addition, we have added a new Table 3 with additional statistical information to clarify the findings.
The authors need to deposit all sequences into GenBank. They state that they only deposited 10% of the sequences. This is unacceptable.
Because of HIV criminalization laws, the standard for depositing HIV sequences is a random 10% of the study population that is anonymized (unlinked to other data) to help maintain the privacy and confidentiality of the participants as we describe in the paper already. This is especially true in densely sampled populations. We now provide a publication to reference this standard. We also note that the authors can be contacted for additional data as appropriate.
Section 3.4.2 and Fig. 3. The authors should label the clusters that are discussed in the text.
We thank the reviewer for this suggestion. We have now placed boxes around the sequences in Fig. 3 with SDRMs so it is now easier to identify the clusters and sequences with SDRMs.
Fig. 1 and associated text. If the authors are going to discuss the largest ‘number’, they should show the data in numbers, not percentages. Also, define as a percentage of what?
We thank the reviewer for this suggestion. We statistically analyzed the SDRMs by year but found these were not significant likely due to the sparseness of the data for each year. We have now removed Figure 1.
Submission Date
12 February 2023
Date of this review09 Mar 2023 09:31:20
Reviewer 2 Report
Alexiev et al. here describe the results of baseline drug resistance analyses on patients newly diagnosed between 2012 and 2020 in Bulgaria. This manuscript is well written but I have some remarks/recommendations:
- It would be of added value if some additional information could be given on the overall HIV prevalence in Bulgaria and how this compares with the neighboring countries? It is mentioned that 2,010 individuals were diagnosed with HIV-1 during 2012-2020. This seems to be a very low prevalence.
- Additional information on the current treatment guidelines in the country may also be of interest. This would allow the reader to make a better estimate on the impact of particular mutations.
- The algorithm that is used for identification of transmitted drug resistance dates from more than 10 years ago. It is essential to reflect a bit on the potential consequences thereof: scoring of mutations that impact drugs no longer in use (already briefly discussed) but also, and more importantly, the possibility of missing mutations that impact newer drugs.
- In the paragraph between lines 353 and 359 on page 7 the clustered occurrence of the L90M mutation is discussed. Maybe I missed something, but I find it very difficult to understand how, in a cluster of 13 members with the L90M mutations, three cluster members are supposed to have attracted their infection abroad. Except when all 3 have visited the same place, the chance that they have been infected with a related virus seems almost impossible. Do these sequences also cluster if codon 90 is removed?
- Please add the total length of the sequences used for phylogenetic analysis.
Author Response

(The authors gave the same response as above.)

Reviewer 3 Report
The authors presented data on drug resistance among patients without ARV experience in Bulgaria for the period 2012-2020. This is important work as drug resistance is currently one of the major barriers to achieving the 95-95-95 targets. This is also a very informative work, since the sample density is very high (more than 50%) and the analyzed period is quite long.
At the same time, according to the reviewer, the authors were not very good at presenting their results. The notes listed below describe the desired changes to the manuscript.
1. It would be nice to see more statistics data in the Introduction section. In particular, it would be useful to include information on the total number of people living with HIV in Bulgaria, the incidence trend over the study period. It is also important to provide information on standards for recommended first-line ART regimens for 2012-2020.
2. In a number of places in manuscript, data on the prevalence of HIVDR are poorly presented and confuse the reader. For example, in section 3.3 (lines 193-194) it is indicated that the prevalence of SDRM is highest among MSM (60%), then among heterosexuals (36.6%) and further among IDUs (3.3%). However, this information is of little use. It would be clearer and more informative here and throughout the text to indicate the prevalence of DR, taking into account the number of people in the cohort, namely MSM - 6.7% (36/534), HETERO - 5.5% (22/401), IDUs - 2.0% (2/101) and add confidence interval. Another example is on lines 205-209. Indeed, the greatest number of mutations was found in subtype B viruses (32/60). However, the DR prevalence in this subtype is below average at 5.0% (32/636). It is much more important that the prevalence of mutations, for example, in CRF12_BF is 62.5% (5/8), and in subtype A1 12.8% (5/39).
3. How can one explain the cyclical growth of the DR, shown in Figure 1? Are these differences in prevalence statistically significant? The manuscript does not indicate the distribution of the analyzed clinical samples by year, so it is not possible to estimate the breadth of the confidence interval. It would be useful to indicate such a confidence interval on the graph (Figure 1) for each annual value of HIVDR prevalence.
4. The Y scale in Figure 2B is probably wrong. Stanford scores are not expressed as a percentage.
5. The conclusion in section 3.3 (lines 234-235) should be corrected and softened. Indeed, the MSM group had the highest number of mutations, but the group size was also the largest. If we calculate the prevalence of mutations in different groups, then it turns out that the difference between the prevalence in the MSM and HETERO groups is not statistically significant.
6. Section 3.4.2 is difficult to understand. It would be useful to add a table in which the first column indicated the analyzed parameter (sex, transmission path, subtype, etc), the second column indicated the number (and %) of clustered sequences, and the third column indicated the number (and %) of clustered sequences with DR mutations, in the fourth - the number (and %) of non-clustered sequences with DR, in the fifth - the significance (p) of differences between the second and third columns. So, it will be visually represented how often sequences with DR are clustered.
7. Lines 362-364. Bad wording. It appears that DR mutations were found in one fifth of PLHIV infected abroad. The reason is the same as above. When talking about a specific cohort, in this case people infected outside the country, it is more correct to discuss the prevalence of mutations in a specific cohort of patients.
8. Lines 367-370. Bad wording. The reason is the same.
Author Response

(The authors gave the same response as above.)

Reviewer 4 Report
This study evaluated the prevalence of circulating HIV-1 drug resistance mutations in Bulgaria from 2012-2020 and compared their findings to previous data generated from 1988-2011. They used a surveillance approach to determine the prevalence of drug resistance mutations (DRMs). The study is significant because the prevalence of DRMS can be as high as 16% in other regions and it is serious enough that drug resistance testing is recommended before HIV+ individuals begin ART.
The authors evaluated HIV-1 pol and protease sequences in 1,053 ART-naïve participants, which was a subset of a larger cohort of 2,010 ART naïve individuals diagnosed with HIV-1 in Bulgaria. Of the 1,053 participants, 60 were found to harbor DRMs, which is 5.7% and is similar to the previous time frame. They also found that circulating HIV-1 is diverse. The majority subtype was B (60%) but they also observed F1, CRF02_AG, A1, CRF12_BF, etc. They found that most DRMs were present in transmission clusters of men who have sexual contact with men. They also reported that while the prevalence is similar to the previous period, there appears to be a shift in the route of dissemination with men who have sex with men now being higher than heterosexual contact or injecting drug use. Overall this is an interesting and informative study; however, there are major concerns with the statistical analyses in which the authors evaluated factors that could be associated with harboring DRMs, and other issues as detailed below.
The statistical comparisons are flawed and should be revised. Table 1 should be SDRM+ vs. SDRM-. The comparison of SDRM+ to the total (SDRM+ and SDRM-) is not rational. All text and analyses related to Table 1 should be revised with an appropriate statistical method.
The authors need to deposit all sequences into Genbank. They state that they only deposited 10% of the sequences. This is unacceptable.
Section 3.4.2 and Fig. 3. The authors should label the clusters that are discussed in the text.
Fig. 1 and associated text. If the authors are going to discuss the largest ‘number’, they should show the data in numbers, not percentages. Also, define as a percentage of what?
Author Response

(The authors gave the same response as above.)

Round 2
Reviewer 4 Report
The authors have made improvements to the manuscript as suggested by the reviewers. However, Table 1 is still labeled as HIV-1 infections with SDRM (n=60) in column 2 and Total number of HIV-1 infections (n=1,053) in column 3. Some of the categories now add up to 1,053 total while others do not, such as route, country, migrants, etc. There should be an explanation if the subcategories are overlapping and how this affects the comparison. Also, the number 851 appears three times in column 3, which may be correct and just a coincidence but should be verified that this is not a copy paste artifact in the data. And the authors should confirm that none of the p values have changed as those as listed as the original p values.
The authors also need to clarify whether they did any correction for multiple tests in Table 1 and/or whether that would be necessary.
Author Response
Response to reviewers:
Reviewer 4
Comments and Suggestions for Authors
The authors have made improvements to the manuscript as suggested by the reviewers. However, Table 1 is still labeled as HIV-1 infections with SDRM (n=60) in column 2 and Total number of HIV-1 infections (n=1,053) in column 3. Some of the categories now add up to 1,053 total while others do not, such as route, country, migrants, etc. There should be an explanation if the subcategories are overlapping and how this affects the comparison. Also, the number 851 appears three times in column 3, which may be correct and just a coincidence but should be verified that this is not a copy paste artifact in the data. And the authors should confirm that none of the p values have changed as those as listed as the original p values.
We thank the reviewer for this comment. We re-ran the analysis and made the changes in the Table 1.
The authors also need to clarify whether they did any correction for multiple tests in Table 1 and/or whether that would be necessary.
Table 1 is a univariate analysis only so corrections were not required. The only corrections that were done were for a new analysis within the subtypes that showed CRF12_BF was associated with SDRMs compared to the reference subtype B. For this analysis we used a Bonferroni correction and the significance level was adjusted as 0.01 (=0.05/5). We now add this information to the methods and results